# IsomiR-eQTL: A Cancer-Specific Expression Quantitative Trait Loci Database of miRNAs and Their Isoforms

**DOI:** 10.3390/ijms232012493

**Published:** 2022-10-18

**Authors:** Afshin Moradi, Paul Whatmore, Samaneh Farashi, Roberto A. Barrero, Jyotsna Batra

**Affiliations:** 1Centre for Genomics and Personalised Health, Queensland University of Technology, Brisbane 4059, Australia; 2Translational Research Institute, Queensland University of Technology, Brisbane 4102, Australia; 3eResearch, Research Infrastructure, Academic Division, Queensland University of Technology, Brisbane 4000, Australia; 4Faculty of Health, School of Biomedical Sciences, Queensland University of Technology, Brisbane 4059, Australia

**Keywords:** miRNA, isomiR, isomiR-eQTL, miR-eQTL, GWAS

## Abstract

The identification of expression quantitative trait loci (eQTL) is an important component in efforts to understand how genetic variants influence disease risk. MicroRNAs (miRNAs) are short noncoding RNA molecules capable of regulating the expression of several genes simultaneously. Recently, several novel isomers of miRNAs (isomiRs) that differ slightly in length and sequence composition compared to their canonical miRNAs have been reported. Here we present isomiR-eQTL, a user-friendly database designed to help researchers find single nucleotide polymorphisms (SNPs) that can impact miRNA (miR-eQTL) and isomiR expression (isomiR-eQTL) in 30 cancer types. The isomiR-eQTL includes a total of 152,671 miR-eQTLs and 2,390,805 isomiR-eQTLs at a false discovery rate (FDR) of 0.05. It also includes 65,733 miR-eQTLs overlapping known cancer-associated loci identified through genome-wide association studies (GWAS). To the best of our knowledge, this is the first study investigating the impact of SNPs on isomiR expression at the genome-wide level. This database may pave the way for researchers toward finding a model for personalised medicine in which miRNAs, isomiRs, and genotypes are utilised.

## 1. Introduction

MicroRNAs (miRNAs) are small noncoding RNAs that regulate gene expression by targeting messenger RNAs (mRNAs) [1,2,3]. The dysregulation of miRNAs has been associated with various types of cancer, such as prostate, colorectal, lung, lymphoma, glioblastoma and osteosarcoma [4]. Specific diagnostic miRNA signatures have been reported for several human cancers through miRNA profiling [2,5]. These miRNA expression signatures may also provide an alternative approach for classifying cancer subtypes. For example, miRNA expression profiles can reveal breast cancer subtypes [6,7,8,9]. Some SNPs which are associated with cancer risk may interfere with the function of miRNAs by changing their binding efficiency and specificity to downstream target genes [10,11]. For example, an SNP rs3746444 in the seed region of mature miR-499 disrupts the PI3K/AKT/GSK signalling pathway and is associated with the increased risk of various cancer types such as cervical squamous cell carcinoma, breast cancer, acute lymphoblastic leukaemia, gallbladder cancer, gastric cancer, squamous cell carcinoma of the head and neck, lung cancer, liver cancer, prostate cancer and colorectal cancer [12]. A miRNA gene can produce multiple miRNA isoforms that can differ in length and sequence composition known as isomiRs [13]. The isomiRs can be variations of a given miRNA mature sequence with additional nucleotides either at their 5′- or 3′-ends [14]. Alterations in the processing of mature miRNA from primary and/or precursor transcripts by Drosha and Dicer enzymes can lead to the addition of nucleotides, particularly at 3′-end yielding the formation of isomiRs [13,15]. IsomiRs can also be generated from the canonical miRNA sequence via RNA editing and the presence of SNPs in the mature miRNA sequence [13,15,16].

The relationship between isomiR expression and cancer progression is less explored than between miRNAs themselves [17]. However, a few studies reported a seminal role of isomiRs in tumourigenesis [18,19]. For example, a higher expression of 5′ isomiR of hsa-miR-140-3p in breast tumour cell lines led to reduced cell viability, cell proliferation and migration [20]. Babapoor et al. discovered that the miR-451a isomiR is associated with an amelanotypic phenotype and has a tumour suppressor effect in melanoma by retarding cell migration and invasion [21]. SNPs within miRNA genes have the potential to impact miRNA biogenesis or alter their target specificity [22,23,24]. For example, miR-196a2-SNP (rs11614913) in the mature miR-196a2 is associated with the enhanced processing of mature miR-196a. Moreover, a study reporting binding assays showed that the rs11614913 can affect the binding of mature hsa-mir-196a2-3p to their target mRNA [24,25]. Fehlmann T. et al. [26] developed a web-based tool (miRMaster) that aligns raw small RNA-seq data onto the human genome and reports the mapping of reads overlapping the miRNA precursor sequences [27]. This study revealed 22,000 potential candidate miRNA precursors with one or two mature forms. Furthermore, it reported isomiRs showing shifted nucleotides in their 5′ and 3′ sites as well as isomiRs with SNPs or mismatches (mis-miRNAs). The functional consequences of germline variants have been revealed by expression quantitative trait loci (eQTL) studies, which have been mainly focused on the regulation of protein-coding genes. A few recent studies have also explored the association of germline variants with the expression of long or short noncoding RNAs in cancer [28,29,30]. On the other hand, some isomiRs have a distinct role in cancers. They can target different genes compared to corresponding canonical miRNA [28,29,30], but there are few studies assessing cancer-associated isomiRs. The growing number of miRNAs and the importance of isomiRs in oncogenesis, suggests a need to extend previous studies to further investigate the effect of SNPs on miRNAs and isomiRs expressions [26]. Previous studies have used The Cancer Genome Atlas (TCGA) dataset to discover tumour-specific miRNA expression quantitative trait loci (miR-eQTLs). For example, Freedman et. al. performed eQTL analyses across five tumour types including breast, colon, kidney, lung and prostate cancers [31]. Li et al. created a database for ncRNA-related eQTLs across 33 cancer types [29]. However, these studies do not encompass isomiRs and their eQTLs modulating their expression. Here, using the miRMaster web-based tool, we performed a genome-wide miR-eQTL and isomiR-eQTL study by utilising genotypes and miRNA/isomiR expression data. We showed that the isomiR-eQTL could be independent of the miR-eQTL, 30 tissue cancer types from the TCGA database. We aimed to reveal the genetic regulation of these newly discovered miRNA and isomiRs in TCGA cancer types. The isomiR-eQTL field is not yet fully explored in terms of drawing associations between miRNAs and diverse cancer types. Hence we anticipate that this database provides informative and valuable information to assess the impact of SNPs on isomiR expression and biogenesis. We have also implemented a public, searchable database for the results (https://data.eresearchqut.net/IsomiR_eQTL/index.html; created on 22 February 2022). This valuable database opens important avenues for future research in understanding the role of dysregulated miRNAs and isomiRs.

## 2. Results

### 2.1. Tumour-Specific miR- and isomiR-eQTL Identification

The generation and processing of the genotype and miRNA-seq data have been detailed in the method section and Figure 1. This study processed a total of 8618 samples ranging from 46 Uterine Carcinosarcoma (UCS) samples to 922 breast invasive carcinoma (BRCA) samples (Table 1). We investigated both canonical miRNAs and isomiRs, including those that carry one mismatch and those that shifted their 5′ and 3′ terminal sites. Both known and novel miRNAs reported by miRMaster and their subgroups were processed to detect miR-eQTL. We identified a total of 152,671 miR-eQTL and 2,390,805 isomiR-eQTL considering an FDR < 0.05 (Table 2) across 30 cancer types. Figure 2 and Figure 3 represent the results of breast and prostate cancer, respectively. The graphs for all cancers are provided in Appendix A. Manhattan plots depict the associated *p*-values of multiple SNPs on autosomal chromosomes for all miR-eQTL (Figure 2A and Figure 3A) and isomiR-eQTL (Figure 2B and Figure 3B). Considering an FDR < 0.5, we found 236,746 and 24,712 total miR-eQTL for BRCA and prostate cancer (PRAD) datasets, respectively.

Density plots show the position of all miR and isomiR-eQTL (FDR < 0.05) in the genome (Figure 2C and Figure 3C). Figure 2D and Figure 3D represent the number of unique miRNA precursors found in BRCA and PRAD studies, respectively. These results indicate that SNPs may impact miRNA processing and isomiR generation.

### 2.2. Identification of miR-eQTL Overlapping GWAS Loci

The integration of the GWAS and eQTL study can help us dissect the genetic mechanism of cancers [32]. We used a collection of GWAS data for various cancers, including bladder, breast, cervical, colon, gastric, head and neck, lung, oral, ovarian, pancreatic, skin, stomach and thyroid cancers [33], to analyse the potential contribution of miR and isomiR-eQTL to cancer risk. By using this list of SNPs, we extracted miR and isomiR-eQTL SNPs that are in linkage disequilibrium (LD) with a cancer-associated SNP identified in the GWAS catalogue, at *r*^2^ > 0.5, LD (*r*^2^) calculated using PLINK 1.9 [34], with the data of the 1000 Genomes phase 3. We identified a total of 5255 tag SNPs related to cancers using the GWAS catalogue; 219,534 SNPs were in LD with these tag SNPs. By mapping miR and isomiR-eQTLs to GWAS SNPs (tag SNP + SNPs in LD with tag SNPs), we identified 65,733 miR and isomiR-eQTLs which overlap with known cancer-associated loci (Figure 2E and Figure 3E for the PRAD and BRCA, respectively).

### 2.3. Some IsomiRs Are Associated with Distinct SNPs Rather Than Their miR Counterparts

We focused on hsa-miR-10b-5p in prostate cancer datasets, given that this miRNA is known to be associated with extracellular vesicles and has been reported as a potential prostate cancer biomarker [35]. Firstly, we performed a spearman correlation between the expression of canonical miR-10a-5p and its five isomiRs. The expression of isomiRs including hsa-miR-10b-5p_0F_-2T_0:T->A (correlation = 0.57, *p*-value = 1.89 × 10^−43^), hsa-miR-10b-5p_0F_-2T_0:T->C (correlation = 0.61, *p*-value=1.71 × 10^−50^) are in moderate (0.5 < r < 0.7) correlation with the expression of the canonical miRNA. There is a strong (r > 0.7) correlation between the expression of canonical miRNA and hsa-miR-10b-5p_23:T->A (correlation = 0.79, *p*-value = 3.1 × 10^−106^). The expression of miR-10b-5p_1F_0T_16:A->G (correlation = 0.32, *p*-value = 5.2 × 10^−13^) showed a weak correlation with the canonical miRNA miR-10a-5p. Therefore, it is expected that most of the identified isomiR-eQTLs exist simply because of the canonical miRNA-eQTL. Next, we employed CAusal Variants Identification in Associated Regions (CAVIAR) [35] to find causal SNPs in the miR-eQTL and isomiR-eQTL locus. CAVIAR quantifies the posterior probability of causality for each variant using a relationship between LD structure and z-score value. We found that the expression of canonical miRNA miR-10a-5p is associated with rs56040758 (CAVIAR score = 0.32), rs62173678 (CAVIAR score = 0.28), and rs62173675 (CAVIAR score = 0.20). The CAVIAR analysis yielded rs62173675 (CAVIAR score of 0.42) as the top-ranked causal-associated SNP with hsa-miR-10b-5p_0F_-2T_0: T->C, which is an isomiR with moderate expression correlation with the canonical miRNA. Additionally, rs59849938 (CAVIAR score = 0.33) and rs10594040 (CAVIAR score = 0.26) also showed association with the isomiR. We identified 24 SNPs associated with hsa-mir-10b_hsa-miR-10b-5p_0F_-2T_0: T->A expression, including rs62173675. The only SNP that is associated with hsa-miR-10b-5p_23:T->A is rs62173678. We hypothesise that the association of rs62173675 and rs62173678 with isomiRs is due to the association of these SNPs with the canonical miRNA. Interestingly, miR-10b-5p_1F_0T_16: A->G showed an association with rs59813559 and rs142523986, however, neither SNP was associated with the canonical miRNA. Furthermore, no expression correlation was observed between miR-10b-5p_1F_0T_16: A->G and the canonical miRNA, suggesting that there are seemingly some exceptions where an isomiR-eQTL association is dependent on nucleotide changes in specific isomiRs.

### 2.4. Web Interface

The web interface has six main sections based on the overlap with miRBase [36]; an overview section, miR-eQTL(miRBase), miR-eQTL (miRMaster), isomiR-eQTL (miRBase), isomiR-eQTL (miRMaster) and GWAS-related miR and isomiR-eQTL. These can be selected from the top banner on each page. The result of miRMaster is categorised into six groups based on the overlap with miRBase [36]. To avoid introducing duplicates derived from the miRBase and miRMaster eQTL analyses, we used all canonical miRNAs identified using miRBase and retained only novel miRNAs found by mirMaster. Therefore, miR/isomiR-eQTLs represented on our webserver using two databases are unique and complementary (Figure 4).

Each section provides 30 searchable datasets, one representing each cancer type. The histogram chart for each section summarises the numbers of miR-eQTLs and isomiR-eQTLs found. The miR-eQTL table displays the SNP ID of the miR-eQTLs, SNP alleles, SNP genomic position (hg19), the minor allele frequency (MAF) of SNPs, average call of the SNPs and the SNP-level quality metric of imputation (Rsq), precursor and miRNA ID, Columns 9–12 represent the outcome of the statistical analysis of miR-eQTL analysis: β-value (effect size estimate), *stat**e* (t-statistic of *t*-test), *p*-value, FDR. Datasets for each cancer type are downloadable from each section in either an excel or CSV format.

## 3. Discussion

Primary precursor miRNAs (pri-miRNA) are transcribed from the miRNA gene and then processed into pre-miRNAs in the nucleus. The pre-miRNAs are transported to the cytoplasm and cleaved into mature miRNAs or isomiRs. IsomiRs can differentiate from their canonical counterpart in targeting different mRNAs [37]. SNPs in the *cis*-regulatory promoter regions of miRNA genes have been shown to regulate the expression of miRNA [35]. Moreover, SNPs in the mature miRNA sequence may potentially change their target specificity [38]. Identification of the miR and isomiR-eQTLs that influence a specific miRNA and isomiR, may help develop better screening, intervention and preventive strategies for people at high cancer risk. In this study, we undertook a genome-wide eQTL analysis to identify the effects of SNPs on miRNA and isomiR expression in the TCGA dataset of cancer tissues, and incorporated this information in the form of a searchable database. This database not only includes all miRNA and isomiR reported in miRBase but also includes miRNA and isomiR that are presented by the miRMaster server [26]. Further, by integrating eQTL data with known cancer-specific GWAS data, we identified miR and isomiR-eQTLs associated with cancer loci. End users can explore miR and isomiR-eQTL, and search through the database to interrogate SNPs that can impact miRNA and isomiRs expression.

Two major cancer types impacting our communities are breast and prostate cancers, thus we present some additional analysis related to these two cancers. We found an isomiR associated with breast cancer that is derived from the hsa-miR-99b-5p locus. This miRNA is reported to suppress liver metastasis in colorectal cancer by downregulating *mTOR* [39]. Additionally, it can target *IGF-1R* in gastric cancer [40] and suppress the *fibroblast growth factor receptor 3* gene in lung cancer [41]. In our study, the miR-eQTL of hsa-miR-99b-5p overlaps with the breast cancer GWAS locus and it may potentially play a tumour suppressor role. One interesting example of prostate cancer is miRNA-26a-5p. This miRNA is associated with patient survival and the migration of cancer cells by inhibiting the cell cycle and triggering apoptosis, consequently exerting an antiproliferative effect in prostate cancer [42]. These cancer-related miR and isomiR-eQTLs are important candidates for follow-up functional validation assays and biomarker discovery.

While our database presents the first iso-miR-eQTL database, there are some limitations to our analysis. Our database is reliant on miRMaster and does not address the caveats associated with this tool. For example, miRNAs with identical sequences that are generated from different regions cannot be discriminated by the mapping of small RNAs. Thus, caution needs to be exercised for identical miRNAs. These loci are typically indexed by 1 to 5 at the end of the precursor name. For example, hsa-mir-941-1, hsa-mir-941-2, hsa-mir-941-3, hsa-mir-941-4, and hsa-mir-941-5. Results for these duplicated mature miRNAs possibly represent the sum of expression of up to five distinct identical loci. In these cases, miR and isomiR association is performed with an expression of these miRNAs with all corresponding loci. We recommend using miR- and isomiR-eQTLs results for single-copy genes or duplicated genes with sequence divergence.

Our results show that some SNPs may impact the transcription of the miRNA genes. These SNPs may be functional variants in cancers that regulate miRNA expression, while other SNPs may specifically impact the isomiR expression. We also observed the expression correlation between miRNA and their isomers. As an example, we assessed the correlation of expression between the canonical miRNAs and isomiRs of hsa-miR-10b-5p using spearman correlation analysis. We found that the expression of its five isomiRs was weak, moderate, or strongly associated with the canonical miRNA.

Further, we generated a Venn diagram depicting the number of unique miRNA precursors found by miR- and isomiR-eQTL approaches for all 30 cancers. In prostate cancer, 32 precursors were common for both miR-eQTLs and isomiR-eQTLs suggesting that SNPs at these loci primarily associate with the canonical miRNA. We also performed a fine-mapping analysis on the hsa-miR-10b-5p eQTL locus and found that some SNPs are associated with both canonical and isomiRs. The expression of these isomiRs is in a moderate to strong correlation with their expression profiles. Interestingly, for SNPs only found to be associated with an isomiR, the expression of the isomiR showed a weak correlation with that of the canonical miRNA. Further fine-mapping analysis for every e-QTL locus is required to discover causal SNPs that impact isomiR expression distinctly.

## 4. Materials and Methods

### 4.1. Genotype Data, Preprocessing and Imputation

Genotype data (Affymetrix 6.0 arrays) from 30 distinct cancer tumour tissues were downloaded from TCGA (https://tcga-data.nci.nih.gov/tcga/ accessed on 26 August 2022). The SNP quality control was performed using genotype data. Individuals with more than 2% missing genotypes were removed. SNPs with call rates < 98% and those SNPs which are not in Hardy–Weinberg equilibrium (HWE) (threshold *p*-value < 10^−8^) were excluded. The SNPs with minor MAFs < 5% were excluded from further analysis. Since a high level of heterozygosity shows low sample quality, whereas low levels of heterozygosity may be due to inbreeding [43], we filtered out the samples with high heterozygosity (mean heterozygosity ± 3 standard deviation (s.d.)) using PLINK 1.9 and R. All samples were checked for sex discrepancies between their recorded sex in the dataset and their genetically determined sex. Genetic studies with individuals of admixed ancestries can confound genetic associations and may increase spurious outputs [44]. For this reason, individuals who were outliers of the European population were removed (6 s.d.). Before the imputation, indels or non-biallelic variants and the ambiguous SNPs that did not match the reference panel, or were duplicates, were excluded using quality control offered by the Michigan server [45]. The genotype data were imputed using the 1000 Genomes Project phase 3 reference panel using the Minimac3 software provided by the Michigan Imputation Server. Analyses were limited to the SNPs with a MAF > 0.5% and a squared correlation coefficient between imputed allele dosages (the sum of the haplotype dosages of each haplotype) and masked genotypes *r*^2^ > 0.5 [45].

### 4.2. Expression Data Processing

RNA-seq bam files of 30 distinct cancer tumour tissues were obtained from the TCGA data portal (https://gdc-portal.nci.nih.gov/ accessed on 26 August 2022). The bam files were sorted and converted to FASTQ format using samtools and bcftools, respectively. The FASTQ files were uploaded to the miRMaster website (www.ccb.uni-saarland.de/mirmaster/ accessed on 26 August 2022), and alignment and quantification were performed by miRMaster. The miRNA gene expression data were generated by miRMaster for each sample. The gene annotation files were downloaded from the miRMaster and miRBase (http://www.mirbase.org/ accessed on 26 August 2022). All miRNAs with an average expression of <1 read per million (RPM) were excluded. Lastly, the expression files were quantile normalized using R.

### 4.3. Identification of miR-eQTLs and isomiR-eQTLs

The Matrix eQTL software (R package) which uses a linear regression model, was utilised for miR-eQTL analysis. The principal component analysis (PC) was performed using PLINK 1.9. Then, PC1 to PC10 were used to assign population stratification. The linear regression model considers the expression values as a dependent variable, and the SNP genotype (imputation dosage) as the independent variable. The analysis was adjusted for PC and gender. The associations were calculated for 30 cancer primary tissue samples. In this analysis, SNPs in the 1 MB distance to miRNA gene on the same chromosome were chosen. Matrix eQTL performs multiple testing corrections using the Bejamini–Hochberg method to estimate the FDR.

### 4.4. Identification of GWAS miR- and isomiR-eQTLs

The list of GWAS SNPs for the available data on 16 cancers, i.e., bladder, breast, cervical, colon, endometrial, gastric, head and neck, kidney, lung, oral, ovarian, pancreatic, prostate, skin, stomach, and thyroid was downloaded from the NHGRI website [33]. GWAS LD regions (1MB distance) were calculated with PLINK 1.9 [34] using the 1000 Genomes phase 3 reference panel. By using this list of SNPs within 1MB regions around GWAS SNPs, we extracted miR-eQTL SNPs that were in LD with a cancer-associated SNP identified in the GWAS catalogue at *r*^2^ > 0.5. Figure 1 depicts the workflow used in this study.

### 4.5. Fine-Mapping

We performed a spearman correlation between the expression of canonical miR-10a-5p and its five isomiRs. We also conducted a fine-mapping approach using CAVIAR software [46] to clarify whether these isomiR-eQTLs relied on the processing of the canonical pre-miRNA.

### 4.6. Database Construction

The miR-eQTL and miR-eQTL-GWAS datasets generated in this study are presented as an online searchable database using R version 4.1.1 (R Developer Core Team, 2021), primarily using the ‘DT’ R package (https://rstudio.github.io/DT/ accessed on 26 August 2022), which provides an R interface for the ‘DataTables’ JavaScript library. These tables are publicly accessible via a web interface at https://data.eresearchqut.net/IsomiR_eQTL/index.html accessed on 26 August 2022, (Figure 4).

## 5. Conclusions

Our miR-eQTL data analysis strengthens the interpretation of genome functionality and provides a valuable framework for the biological understanding of cancer risk. Exploration of the mechanisms of action of the identified miR-eQTLs in this study could lead us to discover networks of dysregulated miRNA and isomiRs contributing to cancers [47,48]. Further studies are required to investigate the functional impact of these miR and isomiR-eQTLs. The user-friendly web interface of isomiR-eQTL provides resources and comprehensive analyses outcomes on the genetic mechanisms of miRNA and isomiR regulation by SNPs in human cancer. We believe that this database is a valuable resource for the research community, particularly in the field of isomiR studies.

## Figures and Tables

**Figure 1 ijms-23-12493-f001:**
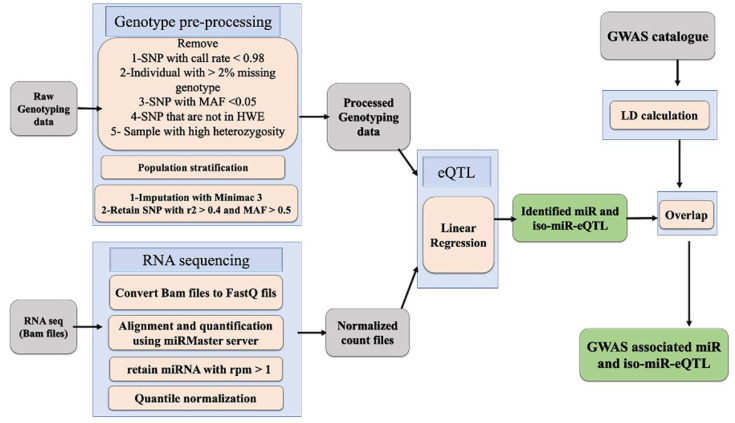
The pipeline of the miRNA and isomiR-eQTL analysis. The grey boxes show the type of data, the blue boxes show the methods of analysis, and the green boxes show the results of this analysis.

**Figure 2 ijms-23-12493-f002:**
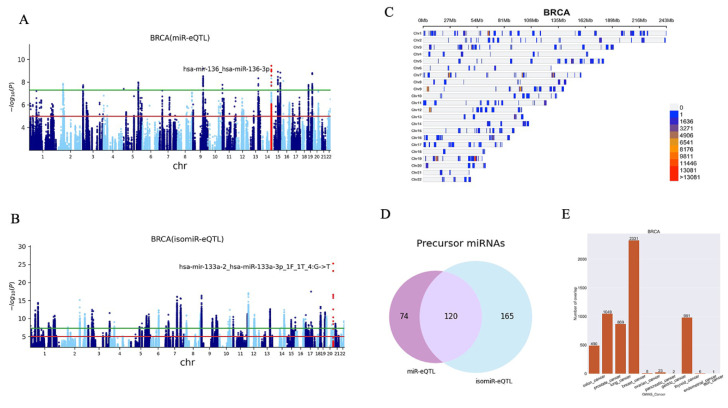
(**A**) The Manhattan plots of miR-eQTL. The most significant miRNA is labelled; it includes the name of the precursor and the miRNA. (**B**) The Manhattan plot of the isomiR-eQTL. The most significant isomiR is labelled; it includes the name of the precursor and the isomiR. (**C**) Density plot shows the position of all miR and isomiR-eQTL (FDR < 0.05) in the genome. (**D**) The Venn graph shows the numbers of miRNA precursors that were discovered in breast cancer samples. (**E**) Bar charts show the number of overlap.

**Figure 3 ijms-23-12493-f003:**
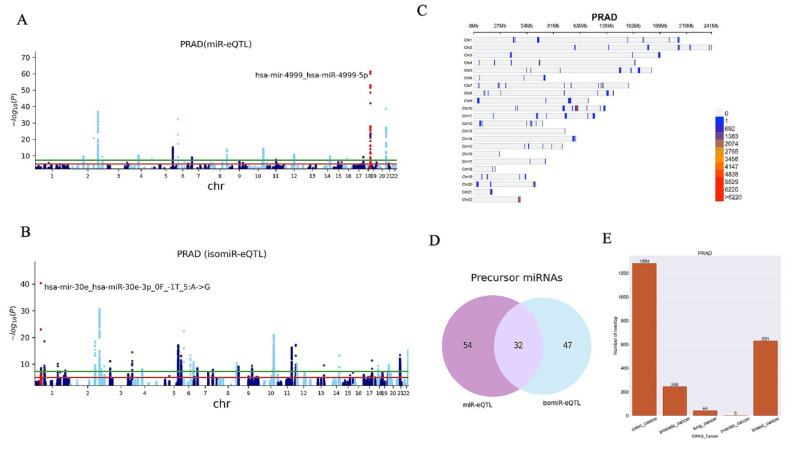
(**A**) The Manhattan plot of miR-eQTL. The most significant miRNA is labelled; it includes the name of the precursor and the miRNA. (**B**) The Manhattan plot of isomiR-eQTL. The most significant isomiR is labelled; it includes the name of the precursor and isomiR. (**C**) Density plot shows the position of all miR and isomiR-eQTL (FDR < 0.05) in the genome. (**D**) The Venn graph shows the numbers of miRNA precursors that were discovered in prostate cancer samples. (**E**) Bar charts show the number of overlap.

**Figure 4 ijms-23-12493-f004:**
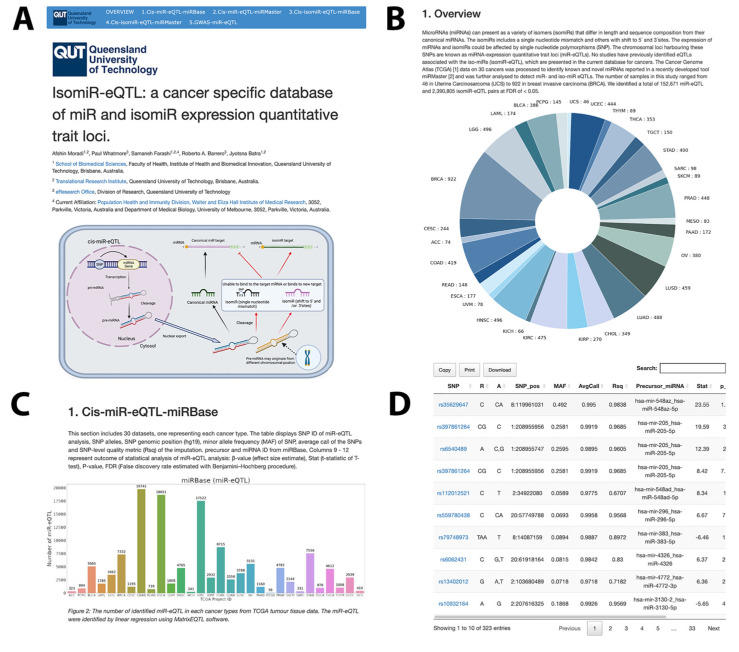
Web interface of the isomiR-eQTL database. (**A**) Home page. (**B**) Pie chart distribution of samples utilised for miR-eQTL analysis. (**C**) Each part includes an overview page that shows the number of identified miR-eQTL in each cancer type from TCGA tumour tissue data. (**D**) miR-eQTL tables include SNPs, alleles, SNP position, miRNA and statistical parameters.

**Table 1 ijms-23-12493-t001:** The number of European samples in each type of cancer in this study. Data for each cancer type were downloaded from (TCGA) (tumour tissue only), where data for genotype and small-RNA-seq was available. The samples of non-European populations were excluded via principal components analysis.

ID	Project	Primary Site	Number of Samples
ACC	Adrenocortical Carcinoma	Adrenal gland	74
PCPG	Pheochromocytoma and Paraganglioma	Adrenal gland	145
BLCA	Bladder Urothelial Carcinoma	Bladder	386
LAML	Acute Myeloid Leukaemia	Haematopoietic and reticuloendothelial systems	174
LGG	Brain Lower Grade Glioma	Brain	496
BRCA	Breast Invasive Carcinoma	Breast	922
CESC	Cervical Squamous Cell Carcinoma and Endocervical Adenocarcinoma	Cervix uteri	244
COAD	Colon Adenocarcinoma	Colon	419
READ	Rectum Adenocarcinoma	Rectum	148
ESCA	Oesophagal Carcinoma	Oesophagus	177
UVM	Uveal Melanoma	Eye and adnexa	78
HNSC	Head and Neck Squamous Cell Carcinoma	Larynx, Hypopharynx, Floor of mouth, and other unspecified parts	496
KICH	Kidney Chromophobe	Kidney	66
KIRC	Kidney Renal Clear Cell Carcinoma	Kidney	475
KIRP	Kidney Renal Papillary Cell Carcinoma	Kidney	270
CHOL	Cholangiocarcinoma	Liver and intrahepatic bile ducts	349
LUAD	Lung Adenocarcinoma	Bronchus and lung	488
LUSD	Lung Squamous Cell Carcinoma	Bronchus and lung	459
OV	Ovarian Serous Cystadenocarcinoma	Ovary	380
PAAD	Pancreatic Adenocarcinoma	Pancreas	172
MESO	Mesothelioma	Heart, mediastinum, and pleura	83
PRAD	Prostate Adenocarcinoma	Prostate	448
SKCM	Skin Cutaneous Melanoma	Skin	89
SARC	Sarcoma	Connective, subcutaneous and other soft tissues	98
STAD	Stomach Adenocarcinoma	Stomach	400
TGCT	Testicular Germ Cell Tumours	Testis	150
THCA	Thyroid Carcinoma	Thyroid gland	353
THYM	Thymoma	Thymus	89
UCEC	Uterine Corpus Endometrial Carcinoma	Corpus uteri	444
UCS	Uterine Carcinosarcoma	Uterus, NOS	46

**Table 2 ijms-23-12493-t002:** The number of identified miR and isomiR-eQTL in each cancer type from TCGA tumour tissue data. The miR and isomiR-eQTL were identified by linear regression using MatrixEQTL software.

ID	Canonical_miRBase	Canonical_miRMaster	isomiR_miRBase	isomiR_miRMaster
ACC	323	194	47	234
PCPG	894	143	897	47
BLCA	5065	1210	29,315	34
LAML	1785	1	2072	63
LGG	3462	206	6276	69
BRCA	7332	3943	216,467	9004
CESC	1195	10	3374	85
COAD	19,741	6002	572,227	18,393
READ	739	3	1751	108
ESCA	18,651	1669	367,816	139
UVM	1808	0	1438	9
HNSC	4765	21	11,807	7
KICH	241	0	492	23
KIRC	17,522	4486	688,332	15,086
KIRP	2932	34	8060	253
CHOL	8715	115	253,732	656
LUAD	2559	20	20,246	41
LUSC	3768	96	8894	617
OV	5531	60	42,409	1248
PAAD	1160	138	471	90
MESO	56	1	591	18
PRAD	4782	130	19,732	68
SKCM	2144	17	752	12
SARC	331	1	688	79
STAD	7556	514	43,171	225
TGCA	978	79	1168	20
THCA	4612	1	3640	4
THYM	1008	1	7224	2
UCEC	2939	517	21,251	498
UCS	410	55	9177	216

## Data Availability

The datasets generated and/or analysed during the current study are available in the isomiR-eQTL repository at: https://data.eresearchqut.net/IsomiR_eQTL/index.html accessed on 26 August 2022.

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
