# Peer review of "IsomiR-eQTL: A Cancer-Specific Expression Quantitative Trait Loci Database of miRNAs and Their Isoforms"

_ijms, 2022, doi:10.3390/ijms232012493_

Round 1
Reviewer 1 Report
This paper addresses an important and interesting problem, they performed a genome-wide miR-eQTL and isomiR-eQTL study by utilizing genotypes and miRNA/isomiR expression data across 30 tissue cancer types from TCGA database. The authors propose to reveal the genetic regulation of these newly discovered miRNA and isomiRs in TCGA cancer types. They antipate this database this database provides informative and valuable information to assess the impact of SNPs on isomiR expression and biogenesis.
Overall, the article is well organized and its presentation is good. However, some minor issues still need to be improved:
(1) There are few explanations of the rationale for the study design.
(2) I suggest that the limitation of this work should be discussed section 3.
(3) The topic is novel but the application proposed is not so novel.
Author Response
Dear,
- There are few explanations of the rationale for the study design.
Response. In agreement with the Reviewer’s comment, we added some explanation for importance of this study in line 71, 72, 86, 90 and 91. Highlighted in yellow
- I suggest that the limitation of this work should be discussed section 3.
Response. Limitation is discussed in lines 251 - 261, we added sentences in lines 258 and 261.
(3) The topic is novel, but the application proposed is not so novel.
Response. We believe eQTL methodology and application is common analysis in genomics study, there are different type eQTL such as sQTL, ncRNA-eQTL, mQTL and the likes. Here we purposed isomiR-eQTL in large scale that is novel type eQTLs comparing to the other types.
Reviewer 2 Report
Your isomiR-eQTL database can be useful tool to support miRNA fields in precision medicine
Author Response
Thank you for review our manuscript. It is much appreciated.
Reviewer 3 Report
In this paper, Moradi et al. have constructed a database of genome-wide isomiR and eQTL. The database is already established and is available to researchers. The paper provides a brief background and overview of the database and should not require major revisions for publication.
minor comments:
1. In Fig.3, legends indicating “A” is laking.
2. Some letters and numbers in Table 2 are difficult to read, so please correct the layout.
Author Response
Dear,
- In Fig.3, legends indicating “A” is laking.
Response. It is corrected
- Some letters and numbers in Table 2 are difficult to read, so please correct the layout.
Response. It is corrected